## PERSPECTIVE

### Calcium waving from the pancreas: the physiological regulation of cytosolic Ca$^{2+}$ signals *in vivo*

**Jason I. E. Bruce** (iD)

*Division of Cancer Sciences, School of Medical Sciences, Faculty of Biology, Medicine and Health, The University of Manchester, Manchester, UK*

Email: jason.bruce@manchester.ac.uk

Handling Editors: Peying Fong & Péter Hegyi

The peer review history is available in the Supporting information section of this article (https://doi.org/10.1113/JP284674#support-information-section).

There have been numerous paradigm-shifting technological advances in the discovery of spatiotemporal Ca$^{2+}$ signalling with pancreatic acinar cells being at the forefront of these discoveries. For many years it was thought that agonist-induced cytosolic Ca$^{2+}$ responses comprised a peak and plateau response. It was not until 1986 that Peter Cobbold and his co-workers first measured agonist-induced cytosolic Ca$^{2+}$ oscillations in single hepatocytes, using microinjected aequorin (Ca$^{2+}$-sensing bioluminescent protein) (Woods et al., 1986). This was a truly paradigm-shifting discovery and led to an explosion of research fully characterizing the complex temporal properties of Ca$^{2+}$ signalling in response to different agonists in multiple cell types.

In acutely isolated pancreatic acinar cells, the hormone cholecystokinin (CCK) induced characteristic slow baseline Ca$^{2+}$ oscillations, whereas the vagally released neurotransmitter acetylcholine (ACh) induced rapid Ca$^{2+}$ oscillations superimposed over a raised baseline (Yule et al., 1991). As the spatial and temporal resolution of imaging microscopy improved, these agonist-induced Ca$^{2+}$ oscillations were shown to be localized to the apical region of pancreatic acinar cells (Thorn et al., 1993). Over the following years there was much debate as to the mechanisms underlying the characteristic ACh- and CCK-evoked Ca$^{2+}$ signals, including differential coupling to cAMP/protein kinase A (and phosphorylation of inositol 1,4,5-trisphosphate receptors) and the second messengers cADP ribose and nicotinic acid adenine dinucleotide phosphate.

Regardless of the precise molecular mechanisms, and despite the huge body of research that went into studying Ca$^{2+}$ signalling in pancreatic acinar cells, the 'elephant in the room' was whether these same phenomena could be observed *in vivo*. How reliable are acutely isolated acinar cells, especially given that they that have been ripped from their natural tissue environment and have no doubt undergone considerable stress and injury? This has been reconciled to some extent with the use of the pancreatic lobule preparation (Won et al., 2011). However, important questions remain. How are these spatiotemporal Ca$^{2+}$ signals, which underlie stimulus–secretion coupling, regulated by physiological stimulation, such as the sight and smell of food, or during feeding, which stimulates CCK release from enteroendocrine cells.

The study by Takano and Yule (2023) in this issue of *The Journal of Physiology* provides another paradigm-shifting technical advance to our understanding of the spatiotemporal Ca$^{2+}$ signalling in pancreatic acinar cells *in vivo*. This is the first study of its kind to fully characterize these Ca$^{2+}$ signals *in vivo* and has far-reaching implications as a platform technology to also investigate how they might be impaired during disease. Using mice specifically expressing the Ca$^{2+}$-sensing fluorescent protein GCaMP6f within acinar cells, they used multiphoton imaging to visualize the specific spatiotemporal Ca$^{2+}$ signals within pancreatic acinar cells of anaesthetized mice *in vivo* in response to physiological stimulation. This included exogenously applied CCK, electrical field stimulation of nerves and feeding *vs.* fasting.

They first showed that pancreatic acinar cells exhibit spontaneous Ca$^{2+}$ oscillations that were inhibited using the CCK antagonist devazepide, but not the muscarinic antagonist atropine, suggesting they are mediated by circulating CCK, rather than the tonic vagal release of ACh. Electrical field stimulated Ca$^{2+}$ oscillations, on the other hand, were inhibited by atropine, suggesting that these are due to vagal release of ACh, as one might expect. Importantly however, both exogenously applied CCK and feeding-induced Ca$^{2+}$ oscillations were blocked by devazepide, but not atropine. This suggests that the feeding-induced response was mediated by the direct action of circulating CCK on pancreatic acinar cells, rather than a vaso-vagal reflex release of ACh, as had previously been suggested.

The study also drilled down at the specific spatiotemporal shaping of these physiologically stimulated Ca$^{2+}$ signals. Both exogenous CCK and feeding produced characteristic 'baseline Ca$^{2+}$ spiking', whereas field stimulated Ca$^{2+}$ oscillations were of higher frequency and sometimes appeared on an elevated baseline, characteristic of ACh-induced Ca$^{2+}$ oscillations observed in isolated acinar cells. The number of responding cells and the frequency of Ca$^{2+}$ oscillations also increased with stimulus strength. Furthermore, low level stimulation produced apically confined Ca$^{2+}$ signals that then propagated as Ca$^{2+}$ waves as stimulus strength increased.

For all of us who have worked painstakingly on acutely isolated pancreatic acinar cells, drilling down at the ever-reductionist molecular details that underlie these spatiotemporal Ca$^{2+}$ signals, we can breathe a collective sigh of relief. Not only does the study by Takano and Yule (2023) provide us with a greater understanding of Ca$^{2+}$ signalling in pancreatic acinar cells within a whole-body context, but it also vindicates the use of acutely isolated acinar cells. This is because the overall spatiotemporal patterns of these Ca$^{2+}$ signals are broadly similar *in vivo* to what they are in isolated cells, thus qualifying previous studies yet providing a springboard for the future. This also provides a platform technology for studying how Ca$^{2+}$ signals become impaired during disease *in vivo*, for example during experimental models of pancreatitis and in genetically modified mouse models of pancreatic cancer. The technology could also be expanded to the *in vivo* imaging of cellular cAMP, ATP (Perceval), the cell cycle (FUCCI) or even cell death (FLIP-GFP). Now that we have 'seen the light' (within pancreatic acinar cells *in vivo*), the horizon is limitless.

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

## Additional information

### Competing interests

The author declares no conflicts of interest.

### Author contributions

Sole author.

### Funding

This work was supported by an MRC grant (MR/P00251X/1) awarded to J.I.E.B.

### Keywords

calcium signalling, *in vivo*, pancreatic acinar cells

### Supporting information

Additional supporting information can be found online in the Supporting Information section at the end of the HTML view of the article. Supporting information files available:

**Peer Review History**

