## [Peer Review History · The Journal of Physiology]

Calcium waving from the pancreas: The physiological regulation of cytosolic Ca²⁺ signals in vivo

Jason I.E. Bruce
DOI: 10.1113/JP284674

Corresponding author(s): Jason Bruce (jason.bruce@manchester.ac.uk)

The following individual(s) involved in review of this submission have agreed to reveal their identity: David I. Yule (Referee #1)

Review Timeline:

Submission Date:	27-Mar-2023
Editorial Decision:	30-Mar-2023
Revision Received:	02-May-2023
Accepted:	10-May-2023

Senior Editor: Peking Fong

Reviewing Editor: Péter Hegyi

Transaction Report:

Dear Dr Bruce,

Re: JP-P-2023-284674 "Calcium waving from the pancreas: The physiological regulation of cytosolic Ca²⁺ signals in vivo"
by Jason I.E. Bruce

Thank you for submitting your manuscript to The Journal of Physiology. It has been assessed by a Reviewing Editor and the authors of the focus paper and we are pleased to tell you that it is acceptable for publication following minor revision.

Please address all the points raised and incorporate all requested revisions or explain in your Response to Referees why a change has not been made. We hope you will find the comments helpful and that you will be able to return your revised manuscript within 2 weeks. If you require longer than this, please contact journal staff: jp@physoc.org.

TRANSPARENT PEER REVIEW POLICY: To improve the transparency of its peer review process, The Journal of Physiology publishes online (as supporting information) the peer review history of all articles accepted for publication. Readers will have access to decision letters, including Editors' comments and referee reports, for each version of the manuscript, as well as any author responses to peer review comments. Referees can decide whether or not they wish to be named on the peer review history document.

REVISION CHECKLIST:

- 'Potential Cover Art' for consideration as the issue's cover image.
- Appropriate Supporting Information (Video, audio or data set: see https://jp.msubmit.net/cgi-bin/main.plex?form_type=display_requirements#supp).

We look forward to receiving your revised submission.

Yours sincerely,

Dr Peking Fong
Senior Editor
The Journal of Physiology
<https://jp.msubmit.net>
<http://jp.physoc.org>
The Physiological Society
Hodgkin Huxley House
30 Farringdon Lane
London, EC1R 3AW

EDITOR COMMENTS

Reviewing Editor:

The inclusion of a summary figure would help to improve understanding and increase the impact of the communication.

Senior Editor:

Thank you for contributing your well-written Perspectives piece.

As you will read, the Reviewing Editor suggests inclusion of a figure. While this would lend a nice touch, I leave this to your judgement.

Please note that the guidelines for Perspectives articles state a limit of 5 references. Therefore, I must ask that you please perform the edits required to meet this requirement.

REFEREE COMMENTS

Referee #1:

We would like to thank Dr Bruce for a thoughtful and balanced perspective on our recent work which nicely puts the study in context of the extensive previous work.

Tiny comment: In the last paragraph Takano is misspelt "Takana"

Responses to Editors/Reviewer – Commissioned Perspectives Article Re: JP-P-2023-284674

"Calcium waving from the pancreas: The physiological regulation of cytosolic Ca²⁺ signals in vivo"

by Jason I.E. Bruce

EDITOR COMMENTS

Reviewing Editor:

The inclusion of a summary figure would help to improve understanding and increase the impact of the communication.

Author Response:

Given that the final version of the article now published online has an excellent summary figure that captures the major strengths and technical advances of this manuscript, I felt that adding another figure would be superfluous and unnecessary.

Senior Editor:

Thank you for contributing your well-written Perspectives piece.

As you will read, the Reviewing Editor suggests inclusion of a figure. While this would lend a nice touch, I leave this to your judgement.

Author Response:

Given that the final version of the article now published online has an excellent summary figure that captures the major strengths and technical advances of this manuscript, I felt that adding another figure would be superfluous and unnecessary.

Please note that the guidelines for Perspectives articles state a limit of 5 references. Therefore, I must ask that you please perform the edits required to meet this requirement.

Author Response:

References have now been reduced.

REFEREE COMMENTS

Referee #1:

We would like to thank Dr Bruce for a thoughtful and balanced perspective on our recent work which nicely puts the study in context of the extensive previous work.

Tiny comment: In the last paragraph Takano is misspelt "Takana"

Author Response:

This minor change has now been corrected.

Dear Dr Bruce,

Re: JP-P-2023-284674R1 "Calcium waving from the pancreas: The physiological regulation of cytosolic Ca²⁺ signals in vivo" by Jason I.E. Bruce

We are pleased to tell you that your paper has been accepted for publication in The Journal of Physiology.

TRANSPARENT PEER REVIEW POLICY: To improve the transparency of its peer review process, The Journal of Physiology publishes online (as supporting information) the peer review history of all articles accepted for publication. Readers will have access to decision letters, including Editors' comments and referee reports, for each version of the manuscript, as well as any author responses to peer review comments. Referees can decide whether or not they wish to be named on the peer review history document.

Authors should note that it is too late at this point to offer corrections prior to proofing. The accepted version will be published online, ahead of the copy edited and typeset version being made available. Major corrections at proof stage, such as changes to figures, will be referred to the Editors for approval before they can be incorporated. Only minor changes, such as to style and consistency, should be made at proof stage. Changes that need to be made after proof stage will usually require a formal correction notice.

Yours sincerely,

Dr Peiyong Fong
Senior Editor
The Journal of Physiology
<https://jp.msubmit.net>
<http://jp.physoc.org>
The Physiological Society
Hodgkin Huxley House
30 Farringdon Lane
London, EC1R 3AW
UK
<http://www.physoc.org>
<http://journals.physoc.org>

P.S. - You can help your research get the attention it deserves! Check out Wiley's free Promotion Guide for best-practice recommendations for promoting your work at www.wileyauthors.com/eeo/guide. You can learn more about Wiley Editing Services which offers professional video, design, and writing services to create shareable video abstracts, infographics, conference posters, lay summaries, and research news stories for your research at www.wileyauthors.com/eeo/promotion.

IMPORTANT NOTICE ABOUT OPEN ACCESS: To assist authors whose funding agencies mandate public access to published research findings sooner than 12 months after publication, The Journal of Physiology allows authors to pay an Open Access (OA) fee to have their papers made freely available immediately on publication.

You can check if your funder or institution has a Wiley Open Access Account here: <https://authorservices.wiley.com/author-resources/Journal-Authors/licensing-and-open-access/open-access/author-compliance-tool.html>.

Editor Comments:

Thank you for contributing this interesting Perspectives piece.

1st Confidential Review

02-May-2023